# p53-Dependent Cytoprotective Mechanisms behind Resistance to Chemo-Radiotherapeutic Agents Used in Cancer Treatment

**DOI:** 10.3390/cancers15133399

**Published:** 2023-06-28

**Authors:** Jayaraman Krishnaraj, Tatsuki Yamamoto, Rieko Ohki

**Affiliations:** Laboratory of Fundamental Oncology, National Cancer Center Research Institute, Tsukiji 5-1-1, Chuo-ku, Tokyo 104-0045, Japan; ram.krishatp@gmail.com (J.K.); tayamamo@ncc.go.jp (T.Y.)

**Keywords:** drug-resistance, DNA damage response, p53, IER5, NRF2

## Abstract

**Simple Summary:**

Acquired resistance to chemoradiotherapy is the common cause of relapse in cancer treatments. Although chemoradiotherapy often produces promising results early in treatment, most cancer patients develop resistance in the later stages and succumb to the disease. In this review, we attempt to explain how cancer cells exploit the tumour suppressor p53 to activate various cytoprotective mechanisms such as DNA damage response, immediate early response gene 5/heat-shock factor 1 pathway, and p21/nuclear factor erythroid 2–related factor 2 pathway to protect themselves from the cytotoxic/genotoxic effects of radiation and drugs. These cytoprotective pathways protect cancer cells by repairing damaged-DNA, maintaining cell homeostasis, reducing oxidative stress, and eliminating drugs from the cells, ultimately resulting in resistance to chemoradiotherapy.

**Abstract:**

Resistance to chemoradiotherapy is the main cause of cancer treatment failure. Cancer cells, especially cancer stem cells, utilize innate cytoprotective mechanisms to protect themselves from the adverse effects of chemoradiotherapy. Here, we describe a few such mechanisms: DNA damage response (DDR), immediate early response gene 5 (IER5)/heat-shock factor 1 (HSF1) pathway, and p21/nuclear factor erythroid 2–related factor 2 (NRF2) pathway, which are regulated by the tumour suppressor p53. Upon DNA damage caused during chemoradiotherapy, p53 is recruited to the sites of DNA damage and activates various DNA repair enzymes including GADD45A, p53R2, DDB2 to repair damaged-DNA in cancer cells. In addition, the p53-IER5-HSF1 pathway protects cancer cells from proteomic stress and maintains cellular proteostasis. Further, the p53-p21-NRF2 pathway induces production of antioxidants and multidrug resistance-associated proteins to protect cancer cells from therapy-induced oxidative stress and to promote effusion of drugs from the cells. This review summarises possible roles of these p53-regulated cytoprotective mechanisms in the resistance to chemoradiotherapy.

## 1. Introduction

Cancer is the leading cause of death worldwide with 10 million cancer related deaths reported in 2020 [1]. A major problem in treating various cancers is the development of resistance to pharmaceutical agents and radiotherapy. In spite of advances in the development of cancer drugs and a better understanding of the molecular mechanisms of cancer, a large number of patients still develops resistance against cancer therapies and succumb to the disease [2]. The tumour suppressor gene *p53* is mutated in approximately half of all human cancers and the mutated *p53* is the primary driving force of tumorigenesis [3]. However, *p53* mutations are rare in certain cancers such as gliomas, neuroendocrine tumours, melanomas, sarcomas, renal cancer, etc. [3,4]. Here, we describe possible mechanisms of resistance against chemoradiotherapy and show how these mechanisms may apply to cancers with wild-type *p53*. We start by describing the physiological mechanisms of p53-mediated DNA damage response (DDR), p53-immediate early gene 5 (IER5)-heat-shock factor 1 (HSF1) pathway, and p53-p21-nuclear factor erythroid 2-related factor 2 (NRF2) pathway, and then describe how these pathways can be exploited by cancer cells to evade radiotherapy and chemotherapy.

## 2. p53

The p53 protein is a transcription factor and tumour suppressor that regulates a large number of genes, the specifics of which depend on cellular conditions [5]. It is called as the guardian of cells or genome, as it is a common gene mutated in the majority of cancers. In response to genotoxic stress, p53 halts cell cycle progression, activates cytoprotective genes including DNA repair enzymes, and induces apoptosis or senescence under extensive DNA damage [6,7]. Structurally, p53 consists of a C-terminal domain, a central DNA-binding domain, and an N-terminal domain which consists of two transactivation domains (TADs) and a proline-rich domain and is required for transcriptional activity [8,9,10]. p53 has 17 phosphorylation sites, 9 of which are located in the TADs [11,12]. Under physiological conditions, p53 is continuously produced and degraded by its negative regulator mouse double minute 2 (MDM2)-induced ubiquitination [13]. Deletion of the first TAD (Δ1stTAD, called p44, p47 or Δ40p53 when expressed naturally) results in the stabilization of p53 by inhibiting its interaction with MDM2. Δ1stTAD induces apoptosis by transactivating proapoptotic proteins p53 binding protein 2 and T-cell intracellular antigen (TIA1) cytotoxic granule associated RNA binding protein like 1 (TIAL1) [14,15]. Δ1stTAD also induces cell cycle arrest at the G2 phase and transactivates various genes like *PLK2*, *PTP4A1*, and *RPS27L* in response to ER stress, causing ER stress-dependent apoptosis [16].

p53 is regulated by various post-tranlational modifications mainly phosphorylation [11]. Phosphorylation of p53 at Thr18 and Ser20 disrupts its interaction with MDM2 thereby stabilizing p53 [17,18,19,20]. Upon DNA damage, p53 undergoes phosphorylation at Ser15 by ataxia-telangiectasia (A-T) mutated (ATM) kinase, AT and Rad3-related (ATR) kinase and DNA protein kinase (DNA-PK) [21,22], Ser9 by casein kinase 1 [23] and DNA-PK [24], Ser20 by checkpoint kinase 1 (CHK1) and 2 (CHK2) [22,25]. The proline-rich domain of p53 contains five PXXP motifs. Proline 72 (p53-72P) is somewhat polymorphic and is sometimes replaced by arginine (p53-72R) [26] which, in lung and breast cancers, correlates with poor clinical outcomes and greater susceptibility to cancer development [27,28]. The N-terminal structures of p53-72P and -72R are different: the 72P variant shows less induction of p21 and more ubiquitination by MDM2 and rapid degradation, whereas the 72R variant undergoes phosphorylation at Ser6/20 and stays stable to induce more p21 [29]. p53 modulates transcription of its target genes by interacting with p53 response elements in their promoters [30]. A large number of *p53* target genes have been reported so far [31]. We have also dissected several *p53* target genes such as *NOXA* [32], *Reprimo* [33], *AEN* (for apoptosis enhancing nuclease) [34], *FUCA1* (for α-1 fucosidase) [35], *IER5* [36], and Pleckstrin Homology (PH) like domain family A (PHLDA) members such as *PHLDA1* and *PHLDA3* [37,38].

## 3. DNA Damage Response

Cellular DNA is constantly under attack by various endogenous metabolites, ionizing radiation, environmental and dietary carcinogens, as well as drugs such as genotoxic anti-cancer agents [39]. Such DNA damage includes base/sugar alterations, DNA-base mismatches, base adducts, sugar base cyclization, DNA protein crosslinking, or intra and inter-strand cross links, all of which result in DNA single- or double-strand breaks (SSBs and DSBs) [40]. While SSBs and other types of DNA damage are relatively less toxic, DSBs are quite dangerous, causing cell death, genomic instability, or carcinogenesis if left unrepaired [41]. DDR involves a broad signalling system that begins seconds after DNA damage and modulates various cellular processes in a concerted, structured manner. It is initiated by DNA damage sensors and signal transducers (ATM and ATR) and effectors (substrates of ATM and ATR) and induces cell cycle checkpoints, activates various DNA repair mechanisms, modulates gene expression, activity and turnover of a wide array of proteins, and thereby affects many aspects of cellular metabolism (Figure 1) [42].

The eukaryotic cell cycle is regulated at three checkpoints: the G1/S, G2/M, and metaphase/anaphase boundaries [43]. DNA damage causes cell cycle arrest at either G1/S transition that prevents entry into S phase or G2/M transition that prevents entry into mitosis [44]. DNA damage activates two important kinases, ATM and ATR, which each phosphorylate various target proteins including two other protein kinases, CHK1 and CHK2. These DNA damage-activated kinases then phosphorylate p53 leading to cell-cycle arrest via transcriptional upregulation of cyclin-dependent kinase inhibitor p21 [45]. There are different types of DNA repair mechanisms depending on the type of DNA damage. SSBs are repaired by direct repair, nucleotide/base-excision repair, mismatch repair, translesion synthesis, etc., while DSBs are repaired by homologous recombination (HR), non-homologous end-joining (NHEJ) or microhomology-mediated end-joining (MMEJ) [46,47]. Cells with severe DNA damage threaten the life of the organism, as it can often lead to cancer and other diseases. Consequently, cells with extensive DNA damage undergo apoptosis either via mitochondrial (intrinsic) pathway or death receptor (extrinsic) pathway depending on the cell conditions. Various components of DDR are known to modulate both pro- and anti-apoptotic proteins in response to DNA damage [48]. This highly complicated DDR is regulated by various proteins, kinases, microRNAs, etc., chief among them are p53 and ATM [49].

ATM, a key player in the DDR, orchestrates various cellular processes including DNA repair, checkpoint activation, apoptosis, senescence, and alterations in chromatin structure, transcription, and pre-mRNA splicing. In response to DSBs, Nijmegen break syndrome 1 (NBS1) of MRE11-RAD50-NBS1 complex recruits ATM to chromatin and stimulates ATM kinase activity. ATM undergoes autophosphorylation at Ser1981, promoting its conversion from an inactive dimer into active monomers [50]. ATM regulates its downstream signalling pathways by phosphorylating several factors including p53, MDM2, homeodomain interacting protein kinase 2 (HIPK2), CHK2, CDC25C, BRCA1, and NBS1 [49]. In response to DNA damage, ATM phosphorylates p53 either directly or indirectly at multiple Ser/Thr residues, chief among them being Ser15, Ser20, and Ser46, leading to stabilization of p53 and induction of cell-cycle arrest, senescence, or apoptosis [49]. Phosphorylation of p53 at Ser15 by CHK2 and Ser20 by ATM leads to induction of p21 which promotes cell cycle arrest while HIPK2-induced phosphorylation at Ser46 induces apoptosis. ATM stabilizes HIPK2 in response to DNA damage by targeting its negative regulator seven in absentia-homolog (Siah)1 for degradation. HIPK2, once released from Siah, binds directly to p53 to mediate p53 Ser46 phosphorylation [51,52]. Active p53 interacts with its transcriptional cofactors and activates various target genes responsible for cell cycle arrest, DNA repair, apoptosis, and senescence. When the DDR ends, WIP1 phosphatase dephosphorylates MDM2 and p53, stopping many of these responses [53].

## 4. Resistance to Chemoradiotherapy in Cancer

Multidrug resistance (MDR) in cancer is caused by various factors including increased DNA repair, altered drug metabolism, reduced drug intake, enhanced drug efflux, tumor microenvironment, tumor heterogeneity, reduced apoptosis [54,55]. Here, we have tried to cover the roles of DNA repair, p53-IER5-HSF1 pathway and p53-p21-Nrf2 pathway in protecting cancers cells from chemoradiotherapy.

### 4.1. Role of p53-mediated DNA Repair in MDR

Defective DNA repair is central to initiation of tumorigenesis: cells with unrepaired DNA damage passes the defective genetic material on to future cells. Defective DNA repair has been observed in a number of cancers including breast, ovarian, prostate cancers, acute myeloid leukemia [56]. A major functions of p53 as a transcription factor is to coordinate various DNA repair mechanisms by regulating gene expression of various components of nucleotide excision repair, mismatch repair and base excision repair [57,58]. Further, p53 also physically interacts with various component of DNA repair pathways including NHEJ, HR, and MMEJ [59]. Recently, p53 has been shown to be directly recruited by poly (ADP-ribose) polymerase (PARP) to DNA damage sites where it helps deciding DNA repair pathways with the help of p53-binding protein 1 (53BP1) and damage specific DNA binding protein 1 (DDB1). 53BP1 recruited by p53 promotes NHEJ over error-prone MMEJ for DSBs while DDB1 favors nucleotide-excision repair for SSBs [60]. While DNA repair initially suppresses tumorigenesis, it is exploited later by well-established tumours for survival against chemoradiotherapy. This has led to the development of DNA-repair targeted therapies using inhibitors for various DNA repair enzymes such as PARP, ATM, ATR, DNA-PK and CHK1 [56,61]. Prolonged treatment of cancer cells with the chemotherapeutic drug cisplatin has been shown to promote DNA repair that can protect the tumour cells from apoptosis [62]. Inhibition of DNA repair mechanisms has also been shown to enhance killing of drug-resistant tumours by cancer drugs such as cisplatin, doxorubicin and palbociclib [63,64,65]. These data suggest that DNA repair plays a vital role in resistance to chemoradiotherapy.

### 4.2. Role of p53/IER5/HSF1 Pathway in MDR

The immediate early genes (IERs) include genes that encode transcriptional factors such as Fos, Jun, Zinc-finger proteins, Myc, secreted cytokines, cytoplasmic proteins, integral membrane proteins, and other genes. IERs are of two types: fast-kinetic IERs whose transcription peaks in 30 min and returns to baseline by two hours, and slow-kinetic IERs whose transcription has a greater lag and persist longer. *IER5*, a novel member of the slow-kinetics IERs, is an intronless gene consisting of 2110 nucleotides that encodes a highly proline-rich nuclear protein of 327 amino acids (33.7 kDa) containing a PEST-like sequence and multiple phosphorylation and O-glycosylation sites [66,67]. IER5 has two nuclear localization sequences (NLS) between amino acids 217–244. These are recognized by importin-α and bind weakly with importin-β, transportin-1 and -2. Amino acids 217-219 (RKR) constitute NLS 1 while amino acids 240-244 (KKPRR) constitute NLS 2. Both NLS 1 and 2 are important for nuclear localization [68]. The *IER5* gene promoter contains two Ets-1 sites and multiple AP-1 and Sp1 sites. Its N-terminal domain consists of 49 amino acids and is 57% identical and 90% similar to the N-terminal domain of IER2 [66,67].


**IER5 is a Target Gene of p53**


In our pervious comprehensive microarray analysis, we found various target genes of p53 using mutants lacking either one or both of p53 TADs and studied their role in cells. *IER5* emerged as one of the top genes activated by p53 in response to oxidative stress [15,36]. We analysed the *IER5* gene promoter region using the ChIP-seq method in the presence or absence of 5-fluorouracil (5-FU) and found two p53 binding sites: RE1 and RE2 (11 and 46 kb downstream of *IER5* gene). Further, we also detected H3K4 tri-methylation surrounded by mono-methylation around the promoter region, indicating that *IER5* is actively transcribed. We found that RE2 is bound more strongly by p53 compared to RE1. RE2 showed greater H3K27 acetylation than RE1, and this was increased further upon 5-FU treatment. Examination of the higher-order chromatin structure of the *IER5* gene revealed that RE2 was located close to the *IER5* gene promoter, consistent with strong p53 interaction with the *IER5* promoter [36].


**IER5 Protects Normal and Cancer Cells from Stress via p53/IER5/HSF1 Axis**


We have reported that upon DNA damage by 5-FU, doxorubicin, γ-radiation, p53-induced IER5 activates HSF1 that stimulates the transcription of heat-shock protein (HSP) family genes to maintain proteostasis. IER5 regulates HSF1 transcription activity via dephosphorylation at key serine and threonine residues including S121, S307, S314, S323, and T367 via PP2A phosphatase while heat shock-induced activation of HSF1 requires phosphorylation at Ser230, 320, and 326. We also have shown that cancer cells require this p53/IER5/HSF1 pathway for proliferation, and found that increased *IER5* expression is associated with poor prognosis of cancer treatment [36]. A detailed depiction of the p53/IER5/HSF1 axis in normal cells and cancer cells is summarised in Figure 2 [69].

*IER5* gene is negatively regulated by RNA polymerase II-associated factor 1 (PAF1), which binds to enhancers downstream of the *IER5* promoter and hinders its transcription [70]. In addition to the p53-induced heat-shock response, *IER5* was also reported to be associated with various other signalling pathways such as the KRAS and mTOR pathways and angiogenesis pathways [71]. *IER5* expression has been shown to be increased in different experimental models in response to DNA damage, heat-shock, γ-irradiation, sleep deprivation, drugs like valproic acid, protein and peptide bound polysaccharides, etc. [72,73,74,75,76,77,78]. Further, increased expression of *IER5* has also been reported in several cancers, especially post-radio/chemotherapy [79,80]. Several studies have reported that *IER5* expression is increased multi-fold by heat-shock or irradiation [36,81,82]. PARP1 stabilises IER5 protein and along with Ku70-IER5 promotes NHEJ [83]. *IER5* also induces G2/M cell cycle arrest by inhibiting CDC25B transcription vis dephosphorylation by PP2A-B55 phosphatase [84,85,86]. Thus, IER5 appears to exert its function in a context-dependent manner involving its binding to various partners, but these diverse functions of IER5 may all contribute to stress resistance in both normal and cancer cells. As a consequence of radiotherapy or chemotherapy, the cytoprotective p53-IER5-HSF1 axis is induced, enabling cancer cells to escape the genotoxic stress induced by these treatments, leading to cancer therapy resistance.

### 4.3. Role of p53-p21 Pathway in MDR

Upon DNA damage, p53 induces DNA damage-induced checkpoint in cell cycle via p21 which is encoded by *CDKN1A* gene, a downstream target of p53. p21 inhibits cell cycle progression by binding to and inhibiting cyclin-dependent kinase-1 and -2 required for cell cycle progression from G1-S phase transition. Further, p21 also inhibits DNA replication by inhibiting proliferating cell nuclear antigen (PCNA) which is required for DNA polymerase activities. In this way, p53-induced p21 not only inhibit cell cycle progression but also regulates PCNA-dependent DNA replication and repair mechanisms [87]. In cancer therapy, p21 has been reported to contribute to cisplatin resistance in ovarian and testicular cancers. Akt-mediated phosphorylation of p21 at Thr145 prevents translocation of p21 from cytosol to nucleus. Cytosolic phosphorylated p21 binds to procaspase-3 and prevents its conversion into caspase-3 thereby blocking apoptosis in cisplatin-treated cells [88,89]. Further, a single nucleotide polymorphism in codon 3 of *p21* gene (Sp21R) which occurs in 10% of chronic lymphocytic leukemia cases has been reported to cause aggressive resistance to infrared radiation treatment [90]. Hypermethylated p21 reduced overall survival rate (at 7 years) of acute lymphoblastic leukemic patients to 8% [91].


**NRF2**


NRF2, a cap ‘n’ collar transcription factor, is known to regulate various aspects of the DDR either directly or indirectly [92]. Our cells have evolved various adaptive mechanisms to defend against oxidative stress: producing antioxidants, phase II xenobiotic metabolizing enzymes, and phase III drug transporters [93,94]. Genes encoding these enzymes share a common antioxidant response element in their promoter regions that is controlled or activated by NRF2 [95,96,97]. Under non-induced conditions, NRF2 is primarily bound to its inhibitor Kelch-like ECH-associated protein (KEAP)1, which mediates polyubiquitination and subsequent degradation of NRF2 by the 26S proteasome, thereby maintaining NRF2 protein at low levels. In response to oxidative stress, KEAP1 is covalently modified leading to reduced ability to promote ubiquitination of NRF2. As a consequence, free NRF2 is translocated to the nucleus where it dimerizes with a musculoaponeurotic fibrosarcoma protein that facilitates the binding of NRF2 to antioxidant response elements. These AREs are found in a wide array of cytoprotective genes (Figure 3) [98,99], and NRF2 activates the transcription of such antioxidant gene products as superoxide dismutase, peroxiredoxin, glutathione peroxidase, glutathione reductase, thioredoxin, catalase, and heme oxygenase-1 [100]. NRF2 regulates all three phases of xenobiotic metabolism: phase I enzymes including aldo-keto reductases, aldehyde dehydrogenases, cytochrome P450, and NAD(P)H: Quinone oxidoreductase 1, phase II enzymes such as glutathione S-transferase and UDP glucuronosyl transferases, and phase III enzymes like ATP-binding cassette transporters and multidrug resistance proteins [99].


**p53/p21/NRF2 Pathway**


Upon DNA damage, p53-induced p21 competes with KEAP1 for NRF2 binding and forms an NRF2-p21 complex [101]. Formation of this complex is one of the first-line defences against oxidative DNA damage. p21 performs both NRF2-induced cytoprotection and cyclin-induced cell cycle arrest separately in a context dependent manner as NRF2 binding motif in p21 overlaps with cyclin binding motif [102]. In addition to activating various cytoprotective genes against oxidative damage, NRF2 also transactivates various DNA repair enzymes such as 53BP1 and RAD51 and promotes expression of anti-apoptotic protein Bcl-XL and inhibits pro-apototic protein BAX [103,104,105]. Further, multidrug resistance proteins induced by p53-p21-NRF2 are key to MDR as they quickly efflux chemotherapeutic drugs out of cancer cells [106].

## 5. Conclusions

With the invention of new technologies, the last century saw advances in our understanding of various cancers and the development of chemoradiotherapy to treat cancer. While we now have better and precise methods to target and kill cancer cells, most patients develop resistance to chemoradiotherapy and succumb to the disease [1,2]. Therefore, a better understanding of the mechanisms behind resistance to chemoradiotherapy is essential to improve treatment strategies. Radiotherapy and the most common chemotherapeutic agents such as alkylating agents and antimetabolites work mainly by inducing DNA damage and oxidative stress [107], conditions that activate p53 to induce cytoprotective mechanisms in the cell. p53 is the cellular gatekeeper to maintain growth and division. Its versatile roles include cell cycle regulation via p21, induction of DNA repair upon damage, or induction of apoptosis if the DNA damage is extensive. It carries out these functions either directly or via transcriptional regulation of its traget genes [108]. Similarly, NRF2 is a master transcription factor that induces various cytoprotective genes including antioxidants, xenobiotic metabolizing enzymes, multidrug resistance proteins, DNA repair enzymes [103,109]. Cancer cells that retain wild-type *p53* exploit these mechanisms to evade therapy and develop resistance. Following chemoradiotherapy, p53-induced DNA repair enzymes help cancer cells to evade DNA damage-induced apoptosis or cell cycle arrest and permit uncontrolled proliferation [110,111]. Further, the p53-IER5-HSF1 axis protects the cancer cells from proteomic stress [76,77,78] while the p53-p21-NRF2 axis protects them from oxidative stress induced by altered metabolic programming [97]. In fact, various studies have reported contradictory role of p53 against its canonical tumor-suppressor and apoptotic properties. Tumors of various grades and types inclduing ovarian, breast, bladder, NSCLC have shown better sensitivity to chemotherapy when their p53 is mutated while their p53 wildtype counterparts showed less or no response to treatment [112,113,114,115,116,117]. These data suggest that tumors with wildtype p53 may be difficult to treat than tumors with mutated p53.

Tumours generally have a subset of cells called cancer stem cells which possess tumorigenic potential during/post-treatment and ultimately result in tumour relapse. Most of the existing treatment methods (including radiotherapy and chemotherapy) aim to reduce the bulk of tumour size rather than eradicate it. This leaves most of the cancer stem cells present intact within tumours. Post-treatment, these residual cancer stem cells initiate rapid cell proliferation and lead to tumour relapse [118]. Cancer cells and especially cancer stem cells exhibit increased expression of various multidrug resistance proteins which mediate drug efflux. In this way, the cancer stem cells not only protect tumours from therapy but also eliminate much of the drug that enters the cytosol. In short, during chemoradiotherapy, cancer cells act in a survival mode utilizing everything at their disposal to protect themselves against the adverse effects of these therapeutic agents (Figure 4). Therefore, the need of the hour in cancer treatment is to supplement existing therapeutic strategies with inhibition or silencing of these cytoprotective mechanisms exploited by cancer cells. Inhibition of the pathways downstream of p53, such as the IER5-HSF1-HSP and p21-NRF2 pathways that protect cancer cells, as adjuncts to existing therapies could provide new opportunities in cancer treatment.

## Figures and Tables

**Figure 1 cancers-15-03399-f001:**
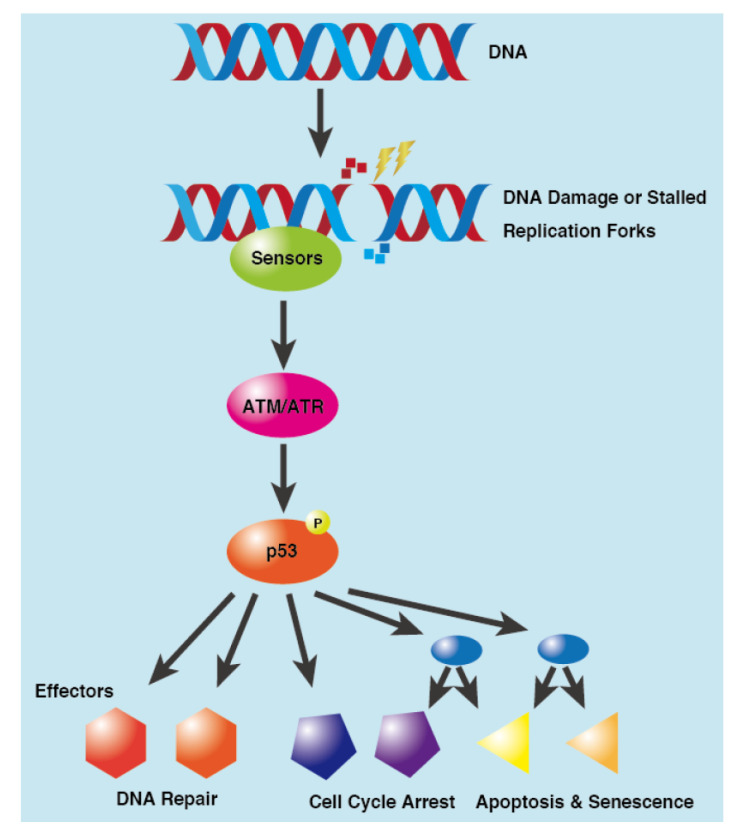
Schematic representation of p53-mediated DNA Damage Response. Upon DNA damage or stalled DNA replication forks, ATM and ATR kinases sense and transduce the signal to downstream effectors that mediate cell cycle arrest, DNA repair, apoptosis, etc.

**Figure 2 cancers-15-03399-f002:**
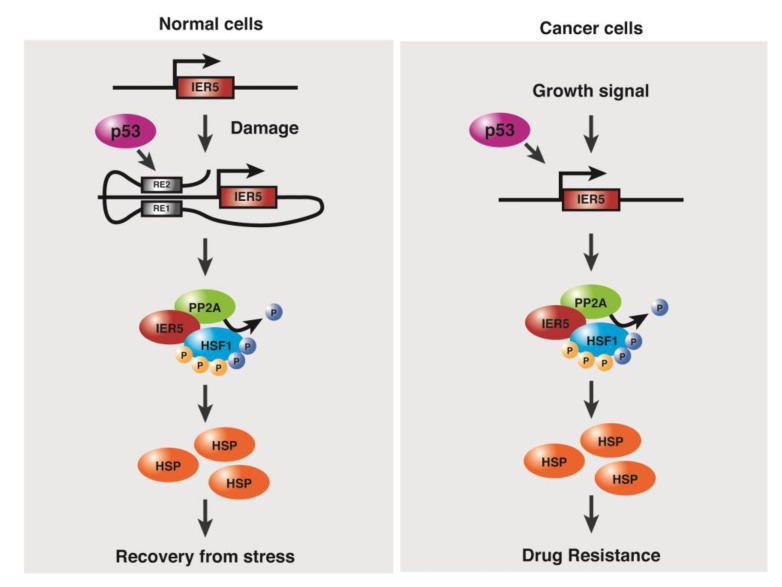
The p53/IER5/HSF1 pathway in normal and cancer cells. Oxidative/genotoxic stress in normal cells induces the p53/IER5/HSF1 pathway, which facilitates recovery from the stress, while cancer cells exploit this innate mechanism to evade therapy.

**Figure 3 cancers-15-03399-f003:**
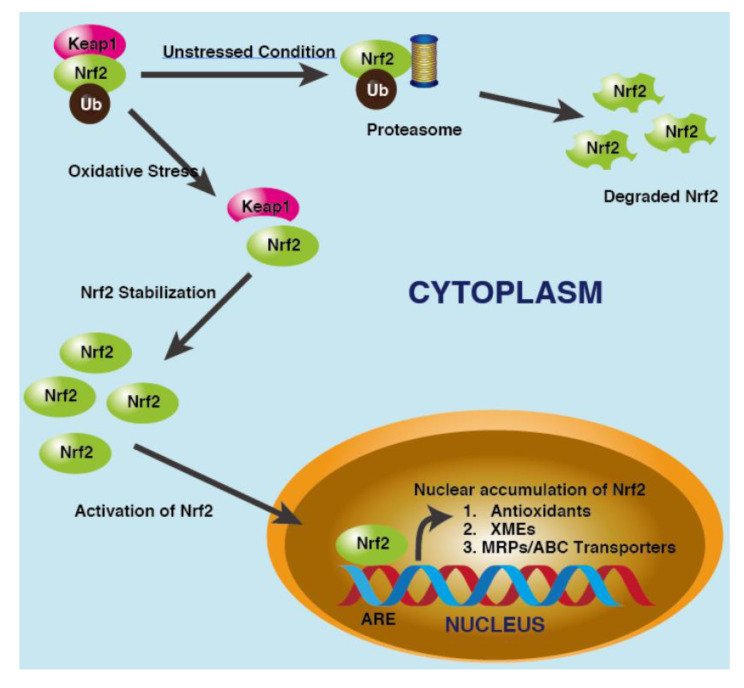
Mechanism of NRF2/KEAP1 signalling. In the absence of cellular stress, KEAP1 promotes polyubiquitination and subsequent proteasomal degradation of NRF2. In response to various stimuli such as oxidative stress, NRF2 is released from KEAP1 and is translocated into the nucleus where it stimulates the transcription of a wide array of cytoprotective genes.

**Figure 4 cancers-15-03399-f004:**
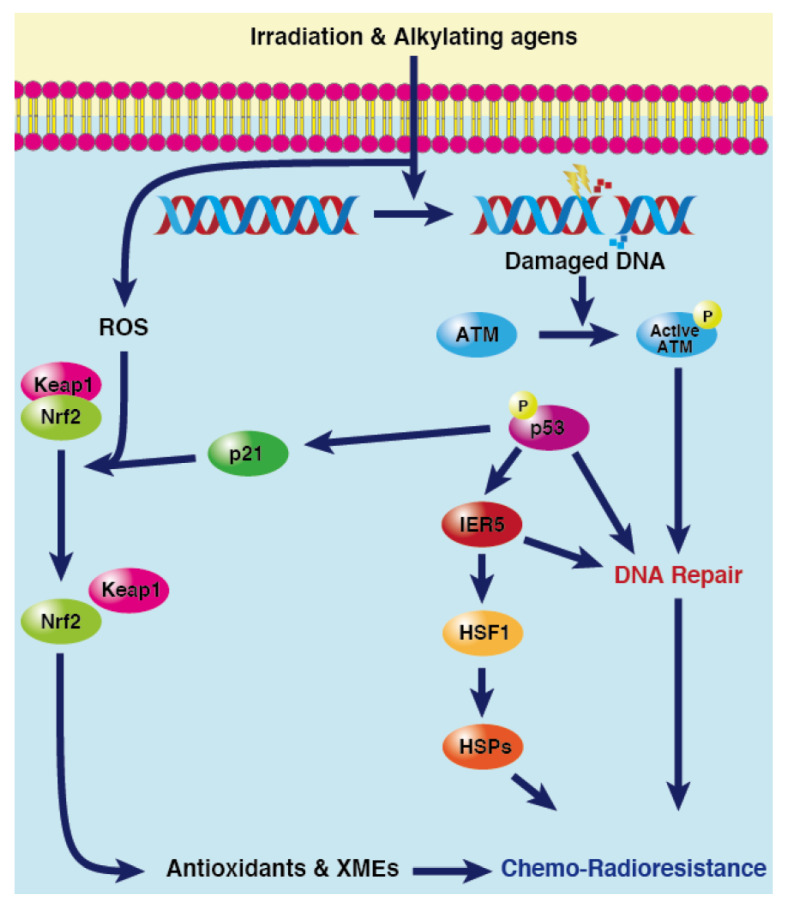
p53-dependent cytoprotective mechanisms underlying resistance to chemoradiotherapy. Upon chemo- or radiotherapy, cancer cells stimulate a wide range of cytoprotective mechanisms such as DDR, IER5/HSF1 pathway, NRF2 signalling, etc., in a p53-dependent manner to evade oxidative/genotoxic stress, thus leading to chemo/radio-resistance.

## Data Availability

No new data were created or analyzed in this study. Data sharing is not applicable to this article.

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
