# Peer review of "p53-Dependent Cytoprotective Mechanisms behind Resistance to Chemo-Radiotherapeutic Agents Used in Cancer Treatment"

_cancers, 2023, doi:10.3390/cancers15133399_

Round 1

Reviewer 1 Report

Krishnaraj et al. present a new point of view about the dangers of p53 action in cancer treatment. This time the culprit is not mutant p53 but wild-type p53. This different side of the p53 story has not been presented or discussed enough so the review is important and timely. I would like to suggest a few ideas for the organization and request some more details:

-       The title is a little misleading as I do not believe p53-IER5, p53-NRF2 or other p53-dependent mechanisms have been targeted in the clinic yet in order to confirm that they actually confer resistance in patients undergoing cancer treatment. Changing the title to “…Behind Resistance to Chemo-Radiotherapy Agents Used in Cancer Treatment” would be more accurate.

-       Also, as the title stands, all p53 cytoprotective mechanisms should be discussed in detail in the review. The authors briefly introduce “DNA repair enzymes [that] accumulate in a p53-dependent manner in cancer cells following radio- or chemotherapy, thus suppressing the induction of DNA damage-induced apoptosis or cell cycle arrest and permitting uncontrolled proliferation” and even draw a third arrow in the main figure (fig. 4) connecting p53 to DNA repair, but they are missing a section dedicated to this important cytoprotective role of p53 (p53 in DNA Repair). A new section should be added about this third axis. Alternatively, the title could be changed to “The roles of IER5 and NRF2 in p53-dependent...".

-       it would be nice if each gene name in the sections’ titles (e.g., 4. IER5; 5. NRF2 SIGNALLING) is accompanied by the name of the pathway/mechanism it affects, in order to have an organization that is more informative and easier to follow

-       The most important section of the article, the one about resistance (presently section 6), which is the main focus within the title and abstract, is only half a page, 10% of the length of the paper. This section(chapter) has to be significantly extended. Here’s a proposal for reorganization that would fix several of the issues mentioned above:

1. Introduction

2. p53

3. DNA damage response

ATM; p53, etc (covering what is now section 2.2.)

4. Resistance to Chemo-radiotherapy

different known mechanisms of resistance, briefly (covering what is now section 2.1.)

4.1 p53-mediated cytoprotection

4.1.1 p53-IER5 axis: heat shock (and/or unfolded protein) response (/ homeostasis?)

4.1.2 p53-NRF2 axis: oxidative stress response

4.1.3 p53-? Axis: DNA repair (new section)

5. Conclusions

OTHER COMMENTS

1.     Figure 1:  Consider changing title of figure 1 to “Schematic representation of p53-mediated DNA Damage Response (DDR)” or include other p53-independent mechanisms as well.

2.     Authors should also discuss the recently discovered role of p53 in direct and transactivation-independent DNA repair (Lane group, PNAS, 2022) (and change figures accordingly)

3.     Only homologous recombination (HR) and non-homologous end-joining (NHEJ) are discussed, MMEJ (microhomology-mediated end joining) should probably also be discussed.

MINOR COMMENTS

1.     Line 44: consider “possible mechanisms”

2.     Line 56: consider dangerous or synonym instead of “lethal”

3.     Line 78: This pathway explains G1 arrest, how about G2 arrest?

4.     Line 84: doesn't have to be irreparable, just leading to too many changes in the sequences/structures of chromosomes.

5.     Lines 88-90: Consider rephrasing. The 2 sentences don't match, the authors were just saying that there's much more, and then they conclude it's all dependent on just p53 and ATM?

6.     Line 93: consider future cells instead of “generation”

7.     Line 95: any example of cancers?

8.     Line 101: resistance or sensitivity?

9.     Line 102: Krishnaraj et al. explain that sensitivity to cisplatin “can be partially restored by inhibition of the NER pathway”, but actually the authors in [18] conclude that it's better to target homologous recombination repair (HRR), like downregulating BRCA2.

10.  Lines 139-140: Consider changing to “Delta1stTAD, called p44, p47 or Delta40p53 when expressed naturally”

11.  Line 143: Consider “resulting in Delta1stTAD accumulation”

12.  Line 147: authors probably mean under normal (physiologic?) conditions; the cells are still normal cells even if they are exposed to stress

13.  Line 175: Consider deleting “position”

14.  Lines 191-192: Not sure what 11 Kb or 42 Kb stand for, location, length, distance?

15.  Line 204: What does HSP stand for? Heat Shock Protein?

16.  Figure 2: there are some circles around the picture, partially cropped. Can the authors remove them?

17.  In figure 2 and related to figure 2: I would prefer "super enhancement" instead of “super enhancer” because the enhancer is a sequence and there is no evidence that there are mutations in cancer cells creating new (“super”) enhancers.

18.  Legend of figure 2: Where it reads “cells exploit this innate mechanism to evade therapy”; the figure shows cancer progression, not resistance to therapy, maybe that's later in figure 4?

19.  Line 219: Consider replacing found with "found evidence"; these are putative enhancers to start with [DOI: 10.1038/ng.3167], not clear what defines them exactly, not clear how they become "super".

20.  Line 225: Reference is missing

21.  Line 235: Should Doi be underlined?

22.  Line 266-268: Authors write that p53's interaction with p21 protein allows formation of a NRF2-p21 complex(!). And that interaction with p21 contains an interaction with KEAP1(!). Please correct.

23.  Lines 328-330: I’m not sure there references say anything about p53

24.  Line 349: Consider “Axes that contribute”

Reviewer 2 Report

The manuscript provides a comprehensive review of literature related to IER5 and NRF2 proteins. To what extent these pathways influence resistance to cancer treatment remains unclear as the cited literature relies mostly on in vitro cellular models. The last section of the manuscript (6. Resistance to chemotherapy) is very general and lacks sufficient detail. The idea that p53 response suppresses DNA damage induced apoptosis and permits uncontrolled proliferation is revolutionary but is not supported by any relevant literature. In the opinion of the reviewer, the manuscript is not suitable for publication in Cancers.

Reviewer 3 Report

Comments: It is true that p53 mutation occur in ~50% of cancers, and some p53 mutants even exhibit gain-of-function effects, which lead to greater drug resistance, so much so it is becoming increasingly evident that resistance is also seen in cancers harboring wild-type p53- not essentially exclusively from mutant p53.  

It would be nice and very informative to the readers to include a paragraph connecting this part of the latest developments and reasonings to consider evidence that the resistance in cancer therapy seen in harboring a wild-type p53 cancers can be substantially greater than that seen in mutant p53 cancers, and this poses a far greater challenge for efforts to design strategies that increase drug response in resistant cancers already primed with wild-type p53. It is noteworthy that specific mutations in p53 not only may result in loss of normal p53 functions but also can induce novel functions, leading to the so-called p53 gain-of-function phenotype, which still confers drug resistance 

Moreover, while connecting p53-p21, it would be nice to include recent developments on mechanistic intervention for the cisplatin resistance in testicular and ovarian cancers is due to the binding of phosphorylated form of p21 in the cytoplasm to procaspase-3, preventing its conversion to caspase-3 and thereby blocking apoptosis.  

Similarly, while p21 gene is rarely mutated, it can also influence therapeutic outcomes through other mechanisms including a single nucleotide polymorphism at codon 31 of p21 in ~10% of chronic lymphocytic leukemia (CLL) patients is sufficient to cause greater aggressiveness and therapeutic resistance or reduced p21 expression through p21 promoter hypermethylation in the corresponding acute lymphocytic leukemia (ALL) leads to a low survival rate of 6–8% in patients compared with a higher rate of ~60% when the promoter is hypomethylated and fully functional.  

The entire review is somewhat hard to comprehend at several paragraphs as outlined in a few incidences below due to the English writing skills that hindered the outcome. I would strongly suggest the review be edited for English grammar and punctuations in order to have a clear and transparent flow of the science. There is much confusion throughout the review, and it is very difficult to grasp. Overall, the scientific structures and goals are well set forth.  

Line 43: Delete (NETs) since they don’t appear again in the txt.  

Line 57-60: Please mention/ introduce a few DDR eukaryotic sensors and effectors 

Line 73-74:   Full name of proteins- ataxia-telangiectasia (A-T) mutated (ATM) kinase and AT and Rad3-related (ATR) kinase: introduce their full name as they appear first in the txt as in line 60. 

Line 75-76: checkpoint kinase 1 (CHK1) and CHK2……..  change to checkpoint kinase 1 (CHK1) and 2 (CHK2).    

Line 80: “translesion” check spelling. 

Line 99Define PARP with full name. 

Line 120-121:  “ATM phosphorylation of p53 also leads to escape from p53 ubiquitination by Mdm2 and stabilization of p53 [22].”    Please change the sentence for a better interpretation and meaning. 

Line 150-151: “ATM kinase phosphorylates p53 at Ser15, a step that appears vital for 150 phosphorylation of p53 at Ser9/20, which in turn recruits p300/CBP, which in turn promotes the transcriptional activity of p53”- confusing sentence. Please change. 

Line 155-157“The N-terminal structures of p53-72P and -72R are different: the 72P variant shows less induction of p21, more ubiquitination by Mdm2 and more rapid degradation…” confusing sentence. Please change/ rewrite. 

Line 177: Change NLS1 to NLS 1. 

Line 204: Define HSP/ HSF1 with full name as it appears first in the in the text rather than later at line 223.  

Line 235Doi et al.- Remove underline. 

Line 244: Delete short forms (XMEs) since it is not mentioned elsewhere/ repeated.  

Line 247: Delete short form (CNC) since it is not mentioned elsewhere/ repeated.  

Line 318: “While we now have soe very good and…..” Please change sentence/ errors. 

The entire review is somewhat hard to comprehend at several paragraphs as outlined in a few incidences below due to the English writing skills that hindered the outcome. I would strongly suggest the review be edited for English grammar and punctuations in order to have a clear and transparent flow of the science. There is much confusion throughout the review, and it is very difficult to grasp. Overall, the scientific structures and goals are well set forth.  

Reviewer 4 Report

The article titled "p53-Dependent Cytoprotective Mechanisms Behind Resistance to Chemo-Radiotherapy in Cancer Treatment" discusses the primary causes of cancer treatment failure, which are resistance to chemotherapy and radiotherapy. The authors focus on the innate cytoprotective mechanisms employed by cancer cells, particularly cancer stem cells, to safeguard themselves against the detrimental effects of these therapies. The review explores several mechanisms regulated by the tumor suppressor protein p53, including the DNA damage response (DDR), the immediate early response gene 5 (IER5)-heat-shock factor 1 (HSF1) axis, and the p21-nuclear factor erythroid 2–related factor 2 (NRF2) axis.

One of the key aspects highlighted in the article is the activation of the p53 pathway by chemotherapy and radiotherapy. This activation leads to the engagement of DNA repair enzymes downstream of p53, such as GADD45A, p53R2, and DDB2, which facilitate the repair of DNA damage caused by chemoradiotherapy. Furthermore, the IER5-HSF1 axis, which operates downstream of p53, contributes to shielding cancer cells from radiotherapy-induced heat shock and maintaining cellular homeostasis. Another significant finding discussed in the review is the role of p53 in inducing the factor p21, which interacts with NRF2 and enhances its activity in response to DNA damage. NRF2, in turn, promotes the production of antioxidants and multidrug resistance-associated proteins (MRPs) that protect cancer cells from therapy-induced oxidative stress and facilitate the efflux of drugs from the cells.

This review provides a comprehensive overview of the p53-regulated cytoprotective mechanisms implicated in the resistance to chemoradiotherapy. The article delves into the intricate network of pathways and proteins involved in protecting cancer cells from the damaging effects of these treatments. By shedding light on these mechanisms, the authors contribute to the understanding of treatment resistance in cancer and potentially pave the way for the development of new strategies to overcome it and should be published.

Author Response

We thank the reviewer very much for his/her careful consideration of our manuscript

Round 2

Reviewer 1 Report

The revision work addressed all issues. Thank you.

Author Response

The authors appreciate the reviewer’s comments.

Reviewer 2 Report

In the revised manuscript authors addressed my major concerns and edited the text that was confusing in the original version. Most importantly, they toned down the statement that the described mechanisms are important for cancer treatment and instead they now say that they affect cell sensitivity to various inhibitors. Overall the text changes improved the manuscript and I do not object the publication. There are still some critical details that should be corrected. P21 does not inhibit checkpoint kinases but cyclin dependent kinases (Line 231). The study by Kandioler-Eckersberger et al. (reference 116) showed that failure of treatment of breast cancer was related to the presence of TP53 gene mutations and thus this citation does not support the author’s statement that tumors show better sensitivity to chemotherapy when p53 is mutated (Line 325).

Author Response

First of all, we thank the reviewer very much for his/her careful consideration of our manuscript; we appreciate the reviewer’s helpful comments and believe that our manuscript has been significantly improved as a result.  The responses are outlined below.

In the revised manuscript authors addressed my major concerns and edited the text that was confusing in the original version. Most importantly, they toned down the statement that the described mechanisms are important for cancer treatment and instead they now say that they affect cell sensitivity to various inhibitors. Overall the text changes improved the manuscript and I do not object the publication. There are still some critical details that should be corrected. P21 does not inhibit checkpoint kinases but cyclin dependent kinases (Line 231).

The authors thank the reviewer for this critical comment. It has been changed to CDK2 and CDK1.

The study by Kandioler-Eckersberger et al. (reference 116) showed that failure of treatment of breast cancer was related to the presence of TP53 gene mutations and thus this citation does not support the author’s statement that tumors show better sensitivity to chemotherapy when p53 is mutated (Line 325).

In the cited study, while clinical response to FEC (fluorouracil, epirubicin, cyclophosphamide) was found to be dependent on wild-type p53, the cytotoxicity of paclitaxel was related to defective p53. The efficiency of paclitaxel during mitosis might be supported by lack of G1 arrest due to p53 deficiency. Therefore, tumors with p53-deficient tumors may show better sensitivity from paclitaxel.

Reviewer 3 Report

NA.

Author Response

(The authors gave the same response as above.)
